# Temporal transcriptomic profiling of pulmonary thromboembolism reveals persistent NETosis- and ferroptosis-associated gene signatures and enhanced thrombolysis with adjunctive DNase I

**Jialun Chen, Lingshan Chao, Siqin Han, Zaixing Jia, Weihua Chen, Zhenwei Liu, Jingwen Li, Xixin Yan**⑩*

The First Department of Pulmonary and Critical Care Medicine, The Second Hospital of Hebei Medical University, Hebei Key Laboratory of Respiratory Critical Care Medicine, Hebei Institute of Respiratory Diseases, Shijiazhuang, Hebei, China

* yanxixin@hebmu.edu.cn

## Abstract

### Objective

To investigate the dynamic pathophysiology of pulmonary thromboembolism (PTE), focusing on the roles of neutrophil extracellular traps (NETs) and ferroptosis, and to evaluate intravenous DNase I as an adjunct to recombinant tissue plasminogen activator (rt-PA).

### Methods

A rabbit autologous thrombus PTE model was established using 62 animals for the time-course and therapeutic-intervention experiments. Disease progression was assessed at days 1, 3, 7, and 14 using histology (collagen volume fraction, CVF), serology (e.g., MPO-DNA, PAI-1, t-PA), and RNA-sequencing. Bioinformatics identified NETs- and ferroptosis-related genes, validated by qRT-PCR. Based on peak NETs activity at day 7, an intervention study (n = 12) compared rt-PA monotherapy versus rt-PA plus intravenous DNase I, with outcomes assessed 7 days post-treatment. In addition, an independent four-group mechanistic validation cohort (Control, PTE, PTE + DNase I, and PTE + ferrostatin-1 [Fer-1]; n = 5 per group) was used to evaluate NETs, TLR9, phospho-p65, total p65, and GPX4 by ELISA and qRT-PCR.

### Results

The PTE model demonstrated progressive pulmonary fibrosis (increasing CVF, P < 0.05) and a sustained hypofibrinolytic state. The NETs marker MPO-DNA peaked at day 7 (P < 0.01). Transcriptomic analysis revealed persistent activation of neutrophil degranulation and iron homeostasis pathways, with early ferroptosis and late collagen metabolism enrichment. Therapeutically, rt-PA plus DNase I was superior

**Data availability statement:** All relevant data are within the paper and its Supporting information files.

**Funding:** The author(s) received no specific funding for this work.

**Competing interests:** The authors have declared that no competing interests exist.

to rt-PA alone, yielding greater improvements in CVF ($P < 0.01$), pulmonary artery acceleration time ($P < 0.01$), oxygen saturation ($P < 0.001$), left ventricular function ($P < 0.05$), and serological markers of endothelial injury and cardiac strain ($P < 0.05$). In the independent validation cohort, PTE increased circulating MPO-DNA complexes, TLR9, phospho-p65 signaling, and the phospho-p65/total p65 ratio while decreasing GPX4; DNase I attenuated upstream MPO-DNA-associated changes in TLR9-NF-kB signaling, whereas Fer-1 more prominently restored GPX4 expression.

## Conclusion

These findings suggest sustained NETs-associated activity and ferroptosis-related processes during PTE progression, identify stage-specific molecular signatures, and support DNase I as a potential adjunct to enhance thrombolysis, while mechanistic relationships require further validation.

## Introduction

Pulmonary thromboembolism (PTE) exhibits a global lifetime cumulative incidence of approximately 8% for venous thromboembolism (VTE) in Western populations, with lower rates in Asian groups [1]. Mortality reaches about 20% within the first year post-event, often due to comorbidities, while survivors face risks of recurrence and chronic complications such as pulmonary hypertension [1]. Diagnosis relies on risk scores (e.g., Wells criteria) to direct D-dimer testing for low-risk exclusion and CT pulmonary angiography for confirmation [2]. Treatment centers on anticoagulation with direct oral agents for stable cases, escalating to thrombolysis or catheter-based interventions in high-risk scenarios, followed by extended therapy based on provocation factors [3]. Despite these advances, PTE management continues to face challenges including diagnostic delays due to nonspecific symptoms and suboptimal outcomes resulting from inadequate personalized treatment strategies across diverse patient populations [4].

The construction of animal models and high-throughput sequencing technologies are now widely used to study disease progression patterns and screen for novel biomarkers. Liu et al. [5] established a rabbit model of acute PTE to investigate dynamic changes in key fibrinolytic factors t-PA and PAI-1. They discovered that the fibrinolytic system undergoes an "early activation followed by late suppression" pattern after PTE, which may represent the key mechanism underlying incomplete thrombus resolution and progression to chronic thromboembolic pulmonary hypertension. Another study by Sun et al. [6] systematically investigated dynamic changes in circRNA and mRNA expression during deep vein thrombosis (DVT) using a rat inferior vena cava stenosis model. Their findings demonstrate that these molecules play dynamic regulatory roles in DVT progression and show promise as early diagnostic biomarkers and therapeutic targets. However, there remains a need to explore novel diagnostic and prognostic biomarkers for different PTE populations to facilitate the development of individualized precision prevention and treatment strategies.

In recent years, neutrophil extracellular traps (NETs) and ferroptosis have been identified as two critical mechanisms of cell death and immune response that are closely associated with thrombus formation. NETs promote thrombosis by capturing blood cells, activating coagulation factor XII, and enhancing fibrin polymerization, thereby accelerating thrombus formation in arterial and venous systems [7]. Ferroptosis exacerbates thrombosis through endothelial damage, release of inflammatory mediators, and creation of a procoagulant state via tissue factor exposure [8]. This study established a rabbit model of PTE to characterize the dynamic features of the pulmonary thrombotic microenvironment at different time points post-embolism. High-throughput sequencing identified NETs-related genes and ferroptosis-related genes associated with thrombus formation, along with potential molecular markers specific to each embolic stage. Based on these findings, specific PTE time points were selected to compare the thrombolytic efficacy of rt-PA monotherapy versus rt-PA combined with DNase I, a NETs-degrading agent. The study aims to elucidate the temporal dynamics of PTE-related factors across different stages and to identify novel biomarkers for diagnosis and treatment, thereby optimizing screening protocols and thrombolytic strategies to improve clinical outcomes and reduce complications.

Mechanistically, NETs are formed when activated neutrophils undergo chromatin decondensation, primarily mediated by peptidylarginine deiminase 4 (PAD4)-catalyzed histone citrullination, followed by the expulsion of web-like structures composed of decondensed DNA decorated with granular proteins such as myeloperoxidase (MPO), neutrophil elastase (NE), and calprotectin (S100A8/A9). These extracellular chromatin networks entrap platelets and red blood cells, activate the contact pathway of coagulation, and provide a scaffold that promotes thrombus growth and resistance to fibrinolysis. Ferroptosis, a distinct form of regulated cell death, is characterized by iron-dependent accumulation of lipid peroxides, resulting from the failure of glutathione peroxidase 4 (GPX4)-mediated antioxidant defense. In the context of thrombosis, emerging evidence suggests that neutrophil-derived iron-binding proteins such as lactoferrin (LTF) and lipocalin-2 (LCN2) may link neutrophil activation to iron dyshomeostasis, potentially creating conditions favorable for ferroptotic cell death in the pulmonary microenvironment. However, the temporal dynamics of these processes and their interplay during PTE progression remain poorly understood.

## Materials and methods

### Reagents and materials

Recombinant tissue plasminogen activator (rt-PA) was purchased from Boehringer Ingelheim Pharma GmbH & Co. KG (Germany). Deoxyribonuclease I (DNase I) was obtained from Ittabio (Beijing, China) and reconstituted in sterile saline according to the manufacturer's instructions. TRIzol reagent was purchased from Simgen (China). The VAHTS Universal V10 RNA-seq Library Prep Kit (Cat: NR616−02, Lot: 7E831C4) was obtained from Vazyme (Nanjing, China). Picro-Sirius Red staining solution (0.1% Sirius Red F3B in saturated picric acid) was supplied by Jiangsu Sumeike Biological Technology Co., Ltd. The Masson's trichrome staining kit was purchased from MDL (Beijing, China). Pentobarbital sodium (3%) was used for anesthesia and euthanasia. Commercial Enzyme-Linked Immunosorbent Assay (ELISA) kits for the quantification of Brain Natriuretic Peptide (BNP), D-dimer, Heat Shock Protein 47 (HSP-47), Myeloperoxidase (MPO), MPO-DNA complexes, cell-free DNA, Plasminogen Activator Inhibitor-1 (PAI-1), tissue Plasminogen Activator (t-PA), Vascular Endothelial Growth Factor (VEGF), von Willebrand Factor (vWF), S100A8/A9, Lipocalin-2 (LCN2), and Lactoferrin (LTF) were all obtained from Jiangsu Sumeike Biological Technology Co., Ltd. All other reagents were of analytical grade unless otherwise specified. Ferrostatin-1 (Fer-1) was used as the ferroptosis inhibitor in the additional mechanistic validation experiment. Commercial ELISA kits for NETs, TLR9, phospho-p65, total p65, and GPX4, together with the corresponding qRT-PCR reagents for TLR9 and GPX4, were used according to the manufacturers' instructions.

### Ethics statement

All animal experiments were approved by the Institutional Animal Care and Use Committee of the second hospital of Hebei Medical University (Approval No. 2024-AE281) and conducted in strict accordance with the Guide for the Care and

Use of Laboratory Animals (National Institutes of Health). Efforts were made to minimize animal suffering and the number of animals used.

## Modeling groups

A total of 62 healthy adult male New Zealand White rabbits (2.5–3.0 kg, from Hebei Medical University) were acclimatized for one week under standard laboratory conditions (22±2°C, 50–60% humidity, 12h light/dark cycle, *ad libitum* access to food/water).

A total of 62 healthy adult male New Zealand White rabbits (2.5–3.0 kg, from Hebei Medical University) were acclimatized for one week under standard laboratory conditions (22±2°C, 50–60% humidity, 12h light/dark cycle, *ad libitum* access to food/water).

Of these, 42 rabbits were used for the establishment of the PTE model and time-course analysis. Six rabbits served as the control group without intervention. Twenty-four rabbits underwent PTE induction without treatment and were sacrificed at 1, 3, 7, and 14 days post-embolization (n=6 per time point) to evaluate dynamic changes in thrombus progression and the pulmonary microenvironment.

## Construction of the autologous pulmonary thromboembolism model

The rabbit PTE model was established using a standardized autologous thrombus embolization technique. Briefly, 15−20 mL of peripheral venous blood was collected from each rabbit's ear vein 24 hours prior to embolization. *In vitro* whole blood was allowed to clot by incubation at 37°C for 30 minutes without any agitation to induce static thrombus formation. Formed thrombi were then rinsed with saline and sectioned into uniform segments (1−2 mm diameter, 5−10 mm length), with a total weight of 0.2–0.3 g per rabbit to ensure consistent embolic burden.

For pulmonary embolism induction, rabbits were anesthetized with 3% pentobarbital sodium (30 mg/kg, IV). A polyethylene catheter (PE-50 tubing) was surgically inserted via the right external jugular vein and advanced into the main pulmonary artery. Catheter placement was confirmed by pressure waveform monitoring. The prepared autologous thrombi were then slowly injected through the catheter. After thrombus delivery, the catheter was withdrawn, the vein ligated, and the incision closed.

Initial model validation and criteria for successful modeling were performed within 30–60 minutes post-embolization. Successful PTE model establishment was defined by the presence of all three criteria: (1) Echocardiographic evidence of thrombi within the pulmonary arteries and signs of acute right ventricular dysfunction (e.g., significant right ventricular dilation or tricuspid regurgitation); (2) A significant decrease in arterial partial pressure of oxygen ($PaO_2$) by at least 20% compared to baseline, or a $PaO_2 < 70$ mmHg, confirmed by arterial blood gas analysis; and (3) Observable clinical signs such as tachypnea, labored breathing, or cyanosis. Only successfully embolized animals meeting these criteria were included in subsequent experiments.

## Histomorphological evaluation

At each designated experimental endpoint, animals were euthanized by an overdose of 3% pentobarbital sodium (150 mg/kg, intravenously), performed under deep anesthesia in accordance with approved animal care protocols. Following euthanasia, lung tissues, meticulously including the pulmonary arteries containing thrombi, were carefully excised. These samples were immediately fixed in 4% paraformaldehyde for 24 hours, dehydrated through a graded ethanol series, embedded in paraffin, and then sectioned at 4 µm thickness. For general morphological assessment, Hematoxylin and Eosin (HE) staining was performed to evaluate thrombus cellularity, the integrity of the fibrin network, and the extent of inflammatory cell infiltration within the thrombus and surrounding pulmonary parenchyma. To evaluate fibrous components and assess thrombus organization and maturation, tissue sections were stained with Picro-Sirius Red (0.1% Sirius Red

F3B in saturated picric acid, Jiangsu Sumeike Biological Technology Co.,Ltd) for 1 hour at room temperature. After staining, sections were rinsed twice with 0.5% acetic acid, dehydrated through a graded ethanol series, cleared in xylene, and mounted with neutral resin. Additionally, parallel sections were stained using a commercial Masson's trichrome kit (MDL, Beijing, China) following the manufacturer's protocol. In brief, sections were sequentially treated with Weigert's hematoxylin to visualize nuclei, Biebrich scarlet-acid fuchsin for cytoplasm and muscle fibers, and aniline blue for collagen fibers, with differentiation steps carried out using phosphomolybdic/phosphotungstic acid solution between stains. Finally, sections were briefly immersed in 1% acetic acid, dehydrated, cleared, and mounted. All stained sections were imaged under a bright-field microscope (Leica DM3000,German). Collagen volume fraction (CVF) was quantified from five randomly selected fields per section at 200× magnification using ImageJ software. For Picro-Sirius Red-stained sections, CVF was defined as the percentage of Sirius Red-positive area (red-stained fibers) relative to the total tissue area. For Masson's trichrome-stained sections, CVF was calculated as the proportion of blue-stained collagen area to the total tissue area.

## Blood oxygen saturation measurement

Arterial blood gas analysis was performed on blood samples drawn anaerobically via puncture of the central ear artery at all specified time points. This direct measurement of the arterial partial pressure of oxygen ($PaO_2$) was utilized to assess the efficiency of pulmonary gas exchange and the degree of hypoxemia resulting from pulmonary embolization, as well as to evaluate the immediate and sustained improvements in oxygenation post-treatment.

## Serological indicator detection

At each scheduled time point for sample collection, peripheral blood samples (approximately 3 mL) were drawn from the ear vein. Blood was collected into EDTA-containing tubes for plasma preparation or plain tubes for serum isolation. Plasma and serum were obtained by centrifugation at 3000 rpm for 15 minutes at 4°C and subsequently stored at −80°C until analysis. A panel of systemic markers was quantified using commercial Enzyme-Linked Immunosorbent Assay (ELISA) kits (Jiangsu Sumeike Biological Technology Co.,Ltd), strictly following the manufacturers' specific protocols. These markers included Brain Natriuretic Peptide (BNP), D-dimer, Heat Shock Protein 47 (HSP-47), Myeloperoxidase (MPO), Plasminogen Activator Inhibitor-1 (PAI-1), tissue Plasminogen Activator (t-PA), Vascular Endothelial Growth Factor (VEGF), and von Willebrand Factor (vWF) concentration (Jiangsu Sumeike Biological Technology Co., Ltd).

## Echocardiographic evaluation

Before and after the treatment, transthoracic echocardiography was performed on conscious, mildly restrained rabbits using a high-resolution ultrasound system (Vinno 6lab,beijng,China) equipped with a 12 MHz linear array probe. Standard M-mode and 2D-mode images were acquired from parasternal long-axis and short-axis views. Key parameters were measured from three consecutive cardiac cycles and averaged. Left Ventricular Ejection Fraction (LVEF) and Left Ventricular Fractional Shortening (LVFS) were calculated to assess left ventricular systolic function. Crucially, Pulmonary Artery Acceleration Time (PAAT) was measured from the pulsed-wave Doppler spectrum of the pulmonary outflow tract, defined as the interval from the onset of flow to peak velocity.

## RNA extraction and transcriptomic sequencing

Whole blood samples (approx. 1.5 mL) were collected from the PTE Model Group and Control Group at 1, 3, 7, and 14 days post-embolization. Total RNA was extracted using TRIzol reagent (Simgen, China). RNA quality (RIN ≥ 7.0) and quantity were assessed via NanoDrop 2000 (Thermo Scientific, USA) and Agilent Bioanalyzer 2100 (Agilent Technologies, USA). RNA libraries were prepared using the VAHTS Universal V10 RNA-seq Library Prep Kit (Cat:NR616–02, Lot:7E831C4, Vazyme) and sequenced on an Illumina NovaSeq 6000 platform (150 bp paired-end reads). Raw reads

underwent quality control (FastQC v0.11.9) and aligned to the rabbit reference genome (Genome assembly mOryCun1.1) using HISAT2 (v2.2.1). Gene expression (FPKM) was quantified via StringTie package (v2.1.4). Differential expression analysis (DEGs) between each PTE time point and the Control Group was performed using the DESeq2 (v1.30.1) package (adjusted $P < 0.05$, $|\log_2 FC| > 1$). EnhancedVolcano (v1.10.0) package generated volcano plots. GO and KEGG enrichment analyses of DEGs were conducted using the clusterProfiler (v3.18.0) package to explore functional changes over time.

### Identification and validation of key genes

Common DEGs across all time points were identified using Venn diagrams. These common DEGs were then intersected with NETs and ferroptosis-related gene lists (curated from GeneCards Human Gene Database, https://www.genecards.org/) to identify persistently dysregulated genes (top150). Subsequently, Random Forest package (v4.7-1.1) was applied at each time point to rank DEGs by their importance in discriminating PTE from Control (thrombus-DEGs). The top 20 thrombus-DEGs from each time point were intersected with time-point-specific upregulated DEGs to identify characteristic DEGs. All intersection visuals were generated using the VennDiagram (v1.7.1) package. Selected key gene mRNA expression (S100A8/9, LCN2, LTF, MGST3, APOL3, OLAH,ATF5, TSPAN15, CYSRT1) in whole blood was validated by quantitative real-time PCR (RT-qPCR) using the 2^−ΔΔCt method (normalized to GAPDH).

### Treatment groups and regimen

Based on transcriptomic sequencing and validation results, animals that were successfully modeled were selected for intervention at an appropriate PTE time point and randomly divided into two groups (n = 6): an rt-PA monotherapy group (1.0 mg/kg, injected via the marginal ear vein) and a combination therapy group receiving recombinant tissue plasminogen activator (rt-PA) (1.0 mg/kg, intravenous injection) plus deoxyribonuclease I (DNase1, 0.3 mg/kg, intravenous). All drug doses were optimized and determined through preliminary experiments. Rt-PA (Boehringer Ingelheim Pharma GmbH&Co,KG,German)and DNase1 were purchased from(Ittabio,Beijing,China) and reconstituted in sterile saline to the working concentration according to the manufacturer's instructions. Histological, cardiopulmonary function, and serological indicators were assessed before treatment and on day 7 after treatment. The improvement rate was calculated as: (Pre-treatment value − Post-treatment value)/ Pre-treatment value.

### Additional pharmacological validation experiment

To strengthen mechanistic support for the proposed NETs-ferroptosis axis, an independent validation cohort was established at day 7 post-PTE, the time point showing peak NETs activity in the time-course experiment. Rabbits were randomly assigned to four groups (n = 5/group): Control, PTE Model, PTE + DNase I, and PTE + ferrostatin-1 (Fer-1). The PTE model was generated as described above. DNase I was administered intravenously using the same dose as in the therapeutic intervention experiment. Fer-1 was used as a pharmacological ferroptosis inhibitor and administered after successful embolization according to the optimized protocol from our preliminary experiments. Peripheral blood was collected at the predefined endpoint, and serum levels of MPO-DNA, TLR9, phospho-p65, total p65, and GPX4 were measured by ELISA. Whole-blood TLR9 and GPX4 mRNA expression was quantified by RT-qPCR and normalized to GAPDH.

### Statistical analysis

All statistical analyses were performed using R software (version 4.2.3). Appropriate statistical methods were selected based on the data type and distribution characteristics. For between-group comparisons, if data followed a normal distribution and homogeneity of variance was satisfied, the Student's *t*-test or *one-way ANOVA* was used; if variances were unequal, the *Welch's t*-test or *Welch's one-way ANOVA* was applied. For continuous variables that did not follow a normal distribution, the *Wilcoxon rank-sum* test (for two groups) or the *Kruskal-Wallis* test (for

multiple groups) was employed. Histomorphological analysis were presented as mean ± standard error of the mean, and between-group comparisons were performed using the *Wilcoxon rank-sum* test. A two-sided *P*-value < 0.05 was considered statistically significant for all hypothesis tests. For serological marker comparisons across multiple time points, Bonferroni correction was applied to adjust for multiple comparisons. The improvement rate for the intervention study was calculated using absolute pre- and post-treatment values, both of which are reported alongside the derived rates. Quantitative data are presented as mean ± SD unless otherwise indicated. For the additional four-group mechanistic validation experiment (Fig 7), one-way ANOVA followed by Dunnett's multiple-comparisons test using the Model group as the reference was performed. All quantitative figures report the number of biologically independent animals in the legends.

## Results

### Histological characterization of the PTE model

The pathological changes in rabbit lung tissues before and after pulmonary embolism were observed using H&E, Masson, and Sirius Red staining (Fig 1A–1C). On day 1 post-embolism, H&E staining revealed widespread thickening of the alveolar walls, structural loss of some alveolar spaces, accompanied by compensatory emphysema in the surrounding alveoli (red arrows). Dilation and congestion of small pulmonary vessels and alveolar wall capillaries were observed (green arrows), along with infiltration of a small number of eosinophils (yellow arrows) and macrophages (blue arrows). Masson staining showed alveolar wall thickening due to congestion, edema, and inflammatory cell infiltration. Within these thickened walls, blue-stained collagen fibers (green arrows) exhibited deeper staining and increased quantity. In some areas, the alveolar structure was disrupted and disorganized, with blue collagen areas appearing continuous and irregular. Collagen fiber aggregation was also noted around blood vessels (red arrows). Sirius Red staining demonstrated thickening, increased density, and deeper staining of red collagen fiber bundles (blue arrows) within the alveolar walls. The red fibrous network showed an irregular thickening trend. In areas of compensatory emphysema, alveolar septa were ruptured, with red collagen fibers appearing reduced or thinner. On day 7 post-embolism, H&E staining indicated further widespread thickening of the alveolar walls and loss of some alveolar structures. Congestion of small pulmonary vessels and capillaries persisted (red arrows), along with mild tissue autolysis (black arrows). Masson staining suggested more pronounced alveolar wall thickening. Blue collagen fibers (green arrows) were distributed diffusely in a reticular pattern within the alveolar septa, covering a broader area compared to the day 1 group. Some regions showed destroyed and fused alveolar structures, replaced by blue fibrous tissue, with noticeably thickened perivascular collagen cuffs (red arrows). Sirius Red staining revealed significant collagen fiber proliferation. A diffusely distributed red fibrous network was observed within the pulmonary interstitium. Proliferated red collagen fiber bundles surrounded blood vessels and filled the thickened alveolar septa, forming continuous red-stained areas (red arrows). On day 14 post-embolism, H&E staining showed persistently widespread alveolar wall thickening, loss of some alveolar spaces, and surrounding compensatory emphysema (red arrows). Dilation and congestion of small pulmonary vessels and alveolar wall capillaries were present (green arrows). Fibrinoid exudates were visible within the alveolar spaces (blue arrows), and local pulmonary tissue exhibited fibrotic changes (yellow arrows), alongside infiltration of a small number of eosinophils (black arrows). Masson staining displayed massive, dense, clump-like deposits of blue collagen fibers (green arrows), with local formation of distinct fibrotic foci. The normal alveolar architecture was destroyed and replaced. Fibrinoid exudates within the alveolar spaces stained red, contrasting with the blue collagen fibers. Sirius Red staining further confirmed extensive collagen fiber proliferation and aggregation. Focal, dense masses of red collagen fibers were observed (red arrows). Fibrinoid exudates in the alveolar spaces appeared pale yellow (green arrows), forming a sharp contrast with the red collagen. As shown in Fig 1D and 1E, the CVF in lung tissues after PTE modeling was significantly higher than that in normal lung tissues. Furthermore, it exhibited a gradual increase with the prolongation of the embolism time (*P* < 0.05). These results collectively indicate the successful establishment of the PTE model.

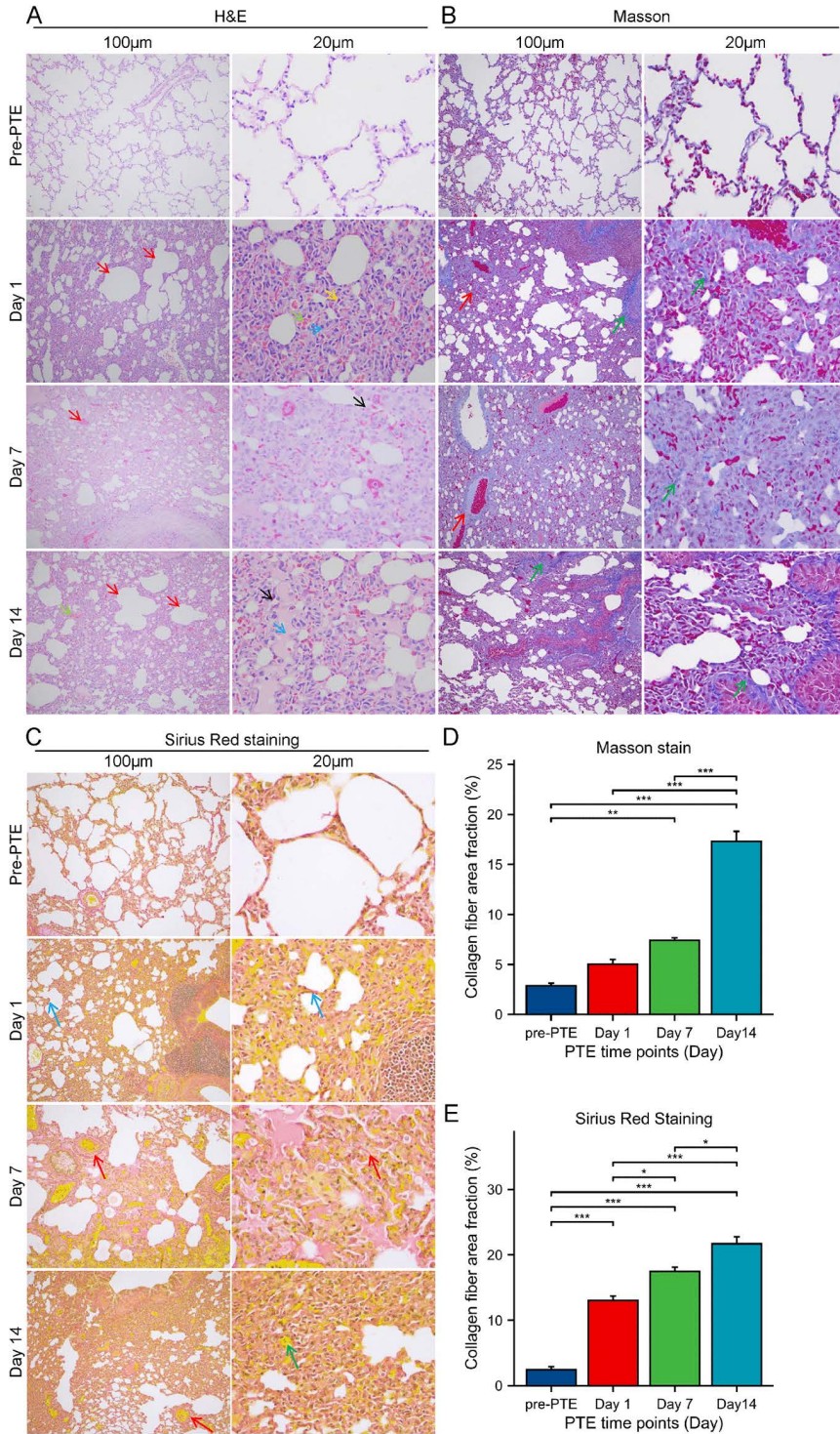

**Fig 1. Histological characterization of the rabbit PTE model.** Representative images of **(A)** H&E, **(B)** Masson and **(C)** Sirius Red staining of lung tissues before modeling and at 1, 7, and 14 days after modeling. Quantitative analysis of the collagen fiber area fraction from **(D)** Masson and **(E)** Sirius Red staining. For the quantitative analyses in **(D)** and **(E)**, n = 6 biologically independent animals per group/time point. *, $P < 0.05$; **, $P < 0.01$; ***, $P < 0.001$. Scale bars: 100 μm and 20 μm.

## Dynamic changes in serological indicators before and after PTE modeling

After PTE modeling, we further examined the changes in serological indicators of the rabbits before and after modeling (Fig 2). Specifically, the serum MPO-DNA level (Fig 2A) increased significantly on day 3 after PTE ($P < 0.05$), peaked on day 7 (compared to before PTE, $P < 0.01$; compared to day 1, $P < 0.05$), and although it decreased slightly by day 14, it remained higher than the pre-PTE level. The serum vWF concentration (Fig 2B) continued to rise after PTE, with levels on days 3, 7, and 14 significantly higher than before PTE ($P < 0.001$). Furthermore, the level on day 14 was significantly higher than at all earlier time points ($P < 0.001$). The BNP level (Fig 2C) peaked on day 3 after PTE ($P < 0.001$), then

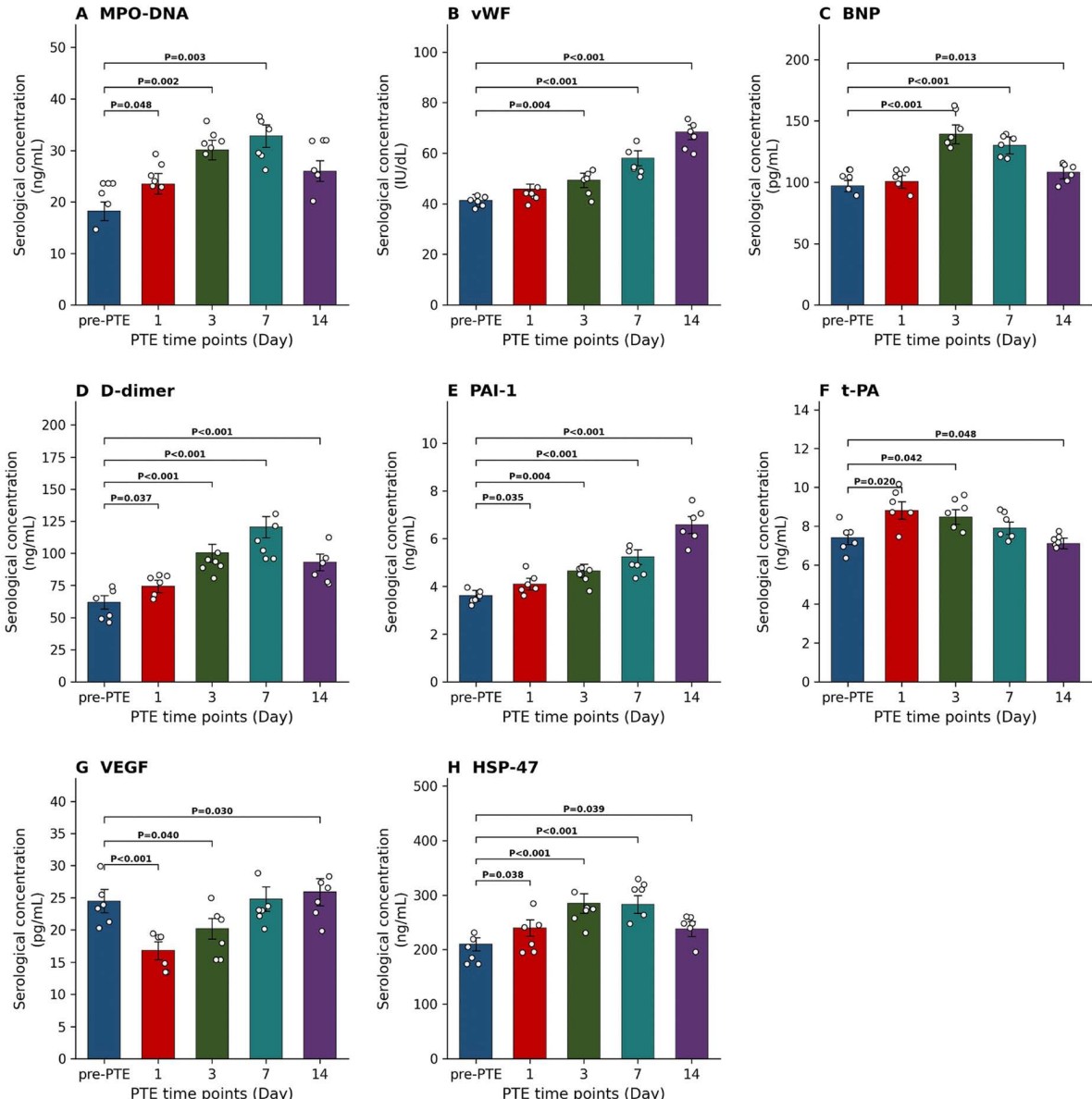

**Fig 2. Dynamic changes in serological indicators before and after PTE modeling. (A)** MPO-DNA. (B) vWF. **(C)** BNP. **(D)** D-dimer. **(E)** PAI-1. **(F)** t-PA. **(G)** VEGF. **(H)** HSP-47. n = 6 biologically independent animals per group/time point. *, $P < 0.05$; **, $P < 0.01$; ***, $P < 0.001$.

gradually declined, but remained significantly higher than the pre-PTE level by day 14 ($P<0.001$). These changes suggest the possibility of sustained inflammatory response, endothelial injury, and cardiac load after PTE.

The serum D-dimer level (Fig 2D) peaked on day 7 after PTE ($P<0.001$), decreased by day 14, but remained significantly higher than the pre-PTE level ($P<0.001$). The PAI-1 concentration (Fig 2E) continued to increase after PTE, with levels on days 3, 7, and 14 significantly higher than before PTE ($P<0.05$). Moreover, the level on day 14 was significantly higher than on day 3 ($P<0.001$). In contrast, the serum t-PA level (Fig 2F) increased significantly on day 1 ($P<0.05$) and then gradually decreased, with levels on days 7 and 14 significantly lower than on day 1 ($P<0.05$). These results indicate that thrombus formation and degradation activities persist after PTE, and the fibrinolytic system may be suppressed.

The VEGF concentration (Fig 2G) decreased significantly on day 1 after PTE ($P<0.001$), then gradually recovered, with the level on day 14 significantly higher than on day 1 and day 3 ($P<0.05$). The serum HSP-47 concentration (Fig 2H) increased significantly and peaked on day 3 ($P<0.05$). This elevated level was maintained until day 7 and decreased by day 14, but remained higher than the pre-PTE level. The changes in these two indicators suggest that the injury repair process after PTE may shift from early angiogenesis to pathological fibrosis.

## Temporal dynamics of DEGs and enrichment analysis following PTE modeling

To identify potential biological markers for PTE, whole blood samples were collected from both the PTE model and control groups at 1, 3, 7, and 14 days post-embolization for transcriptome sequencing. The DEGs at each time point, along with their GO and KEGG enrichment results, are shown in Fig 3. On day 1, the model group exhibited 861 upregulated and 419 downregulated genes compared to the control group. GO and KEGG enrichment analyses were performed on upregulated DEGs using the full set of expressed genes (FPKM > 1 in at least one sample) as background. The upregulated DEGs were primarily enriched in Biological Process (BP) terms such as response to virus, regulation of innate immune response, iron ion homeostasis, heme metabolic process, cellular iron ion homeostasis, transition metal ion transport, secretory granule lumen, ubiquitin ligase complex, specific/tertiary/azurophil granules, endocytic vesicle lumen, proteasome, mitophagy/autophagy, ferroptosis, and porphyrin metabolism (Fig 3A). At day 3, 358 genes were upregulated and 173 were downregulated in the model group, and these upregulated genes were mainly enriched in activation of innate immune response, iron ion homeostasis, heme biosynthetic process, tetrapyrrole biosynthetic process, response to iron ion, cytoplasmic vesicle lumen, secretory granule lumen, specific/tertiary granule lumen, primary lysosome, azurophil granule, endocytic vesicle lumen, and proteasome (Fig 3B). By day 7, the model group demonstrated 327 upregulated and 366 downregulated genes, which were predominantly enriched in pigment granule organization, porphyrin-containing compound biosynthetic process, heme biosynthetic process, tetrapyrrole biosynthetic process, basal part of cell, basolateral plasma membrane, specific granule, tertiary granule lumen, endocytic vesicle lumen, and azurophil granule (Fig 3C). On day 14, 774 upregulated and 1768 downregulated genes were identified, with upregulated genes enrichments primarily in leukocyte migration, external encapsulating structure organization, extracellular structure organization, anatomical structure homeostasis, tissue remodeling, tissue homeostasis, collagen metabolic process, iron ion transport, collagen-containing extracellular matrix, endopeptidase regulator activity, basement membrane, specific granule, blood microparticle, tertiary granule lumen, collagen trimer, cell cycle, and complement and coagulation cascades (Fig 3D). Notably, the granule-related cellular components and iron transport-related terms were consistently enriched across all four time points.

## Identification and validation of common NETs/Ferroptosis and temporal-specific DEGs Post-PTE

The intersection of upregulated DEGs across all time points yielded 46 common DEGs (Fig 4A). Further gene set intersection analysis revealed that during the post-PTE course, LCN2 and LTF were common DEGs associated with both NETs formation and ferroptosis, while S100A8 and S100A9, as NETs-related common DEGs, also demonstrated sustained upregulation (Fig 4B and 4C). We then employed the Random Forest algorithm to identify thrombus-associated DEGs relevant to PTE development at each time point. The intersection of top20 thrombus-associated DEGs (Supporting

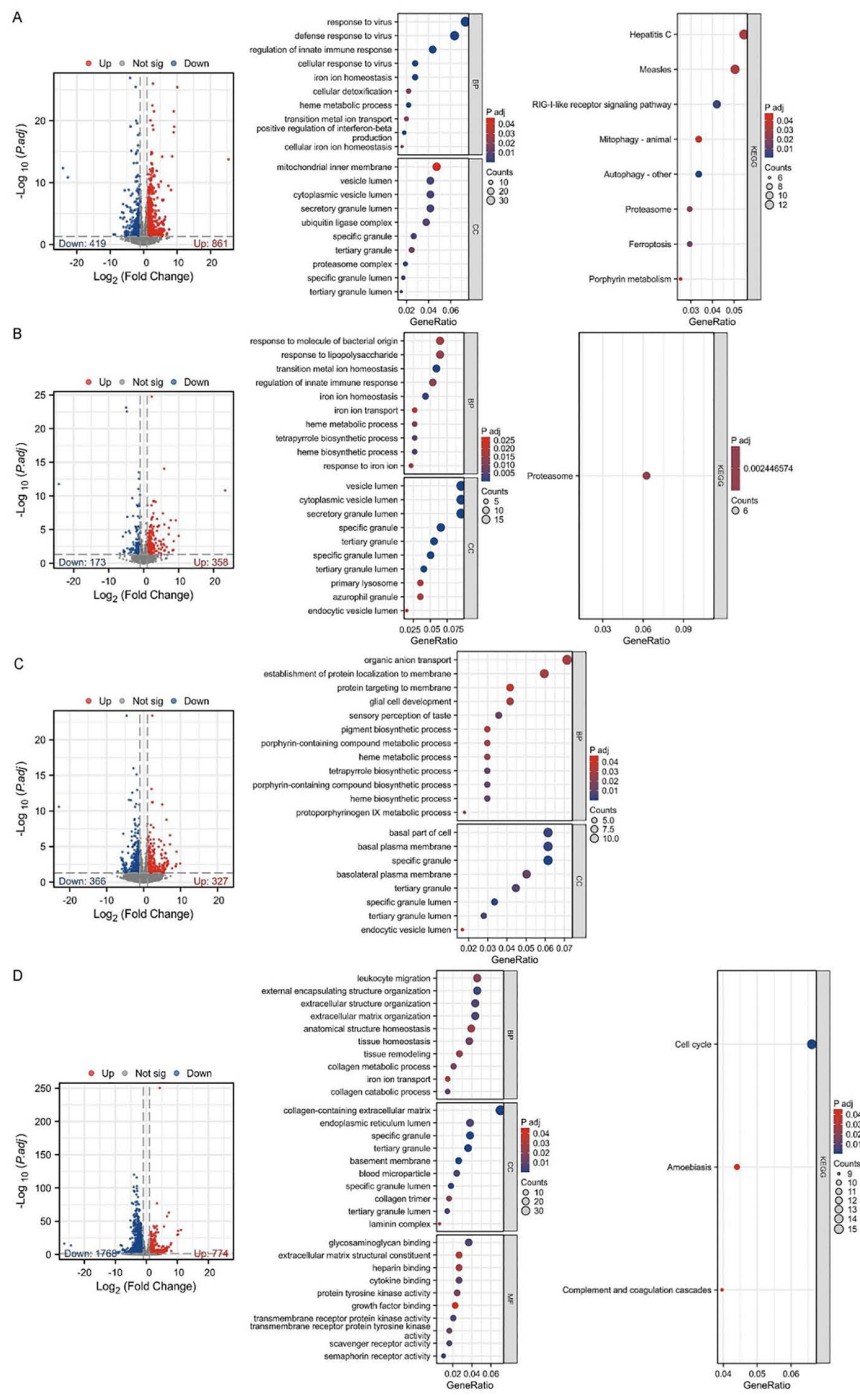

**Fig 3. DEGs and their GO and KEGG enrichment analysis in the PTE model group versus the control group at 1 (A), 3 (B), 7 (C), and 14 (D) days post-modeling.** Whole-blood RNA-seq was performed using n = 6 biologically independent animals per group/time point.

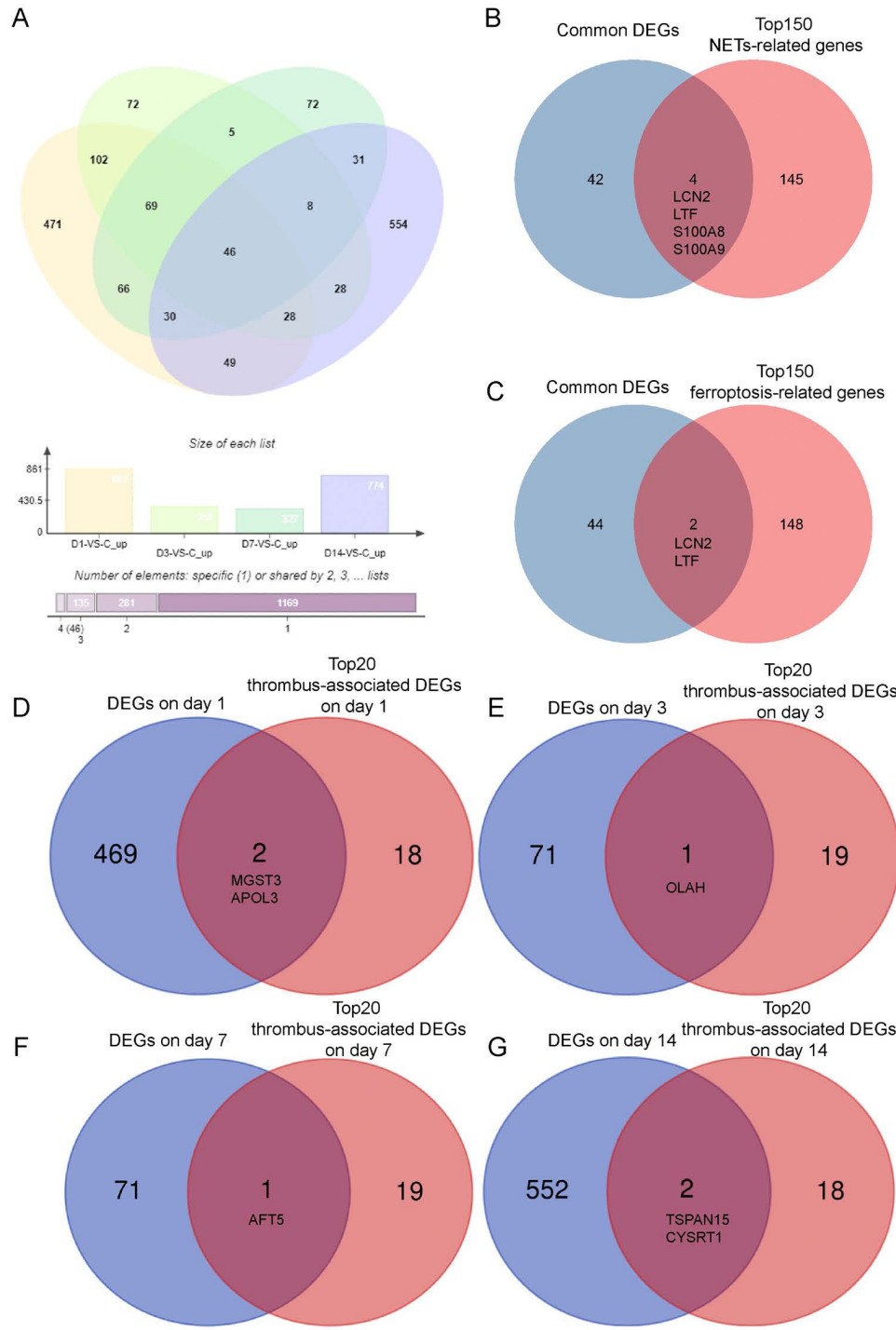

**Fig 4. Identification of common NETs/Ferroptosis and time point-specific upregulated DEGs following PTE. (A)** Venn diagram illustrating the common DEGs obtained from the intersection of upregulated DEGs across all post-PTE time points. **(B, C)** Venn diagrams showing the intersection of the common DEGs with gene sets for NETs **(B)** and ferroptosis **(C)** from the GeneCards database. **(D-G)** Venn diagrams identifying key time point-specific thrombus-associated DEGs at day 1 **(D)**, day 3 **(E)**, day 7 **(F)**, and day 14 **(G)** post-embolization, revealed by intersecting the top 20 thrombus-associated DEGs with the time point-specific upregulated DEGs. RNA-seq input samples were derived from n = 6 biologically independent animals per group/time point.

Information) with the time point-specific upregulated DEGs revealed distinct key genes for each stage: MGST3 and APOL3 at day 1, OLAH at day 3, ATF5 at day 7, and TSPAN15 and CYSRT1 at day 14 (Fig 4D–4G).

To evaluate the potential of the aforementioned commonly and time-specifically identified genes as biomarkers for PTE, we performed qRT-PCR analysis. The results confirmed that the shared NETs/ferroptosis-related genes LCN2, LTF, S100A8, and S100A9 were significantly upregulated at all post-embolism time points compared to pre-PTE levels (Fig 5A). Specifically, LCN2 expression increased markedly ($P<0.001$), peaking at day 7, suggesting its sustained role in inflammation and iron transport. LTF was significantly elevated throughout the 14-day period ($P<0.05$ at days 1 and 3; $P<0.01$ at days 7 and 14), with further increases at days 7 and 14 compared to day 1 ($P<0.01$), indicating persistent iron dyshomeostasis and inflammatory activity. Both S100A8 and S100A9 showed substantial upregulation ($P<0.001$ and $P<0.01$–$0.001$, respectively), with peaks at day 7 and day 3, respectively, supporting their roles as inflammatory mediators in neutrophil/monocyte-driven responses. For time point-specific DEGs (Fig 5B–5E), MGST3 and APOL3 were significantly elevated at day 1 (Fig 5B, $P<0.001$), with MGST3 remaining upregulated through day 14 ($P<0.05$), reflecting early oxidative stress and detoxification demands. OLAH peaked at day 3 (Fig 5C, $P<0.001$), indicating a role in acute-phase lipid metabolism. ATF5 expression rose significantly by day 7 (Fig 5D, $P<0.01$ vs. pre-PTE, day 1, and day 3), implying involvement in cellular stress and repair processes during the subacute phase. Finally, TSPAN15 and CYSRT1 exhibited progressive upregulation, becoming significant from day 7 ($P<0.05$) and peaking at day 14 (Fig 5E, $P<0.001$ and $P<0.01$, respectively), suggesting their potential roles in chronic vascular remodeling and tissue repair. In summary, qRT-PCR validation corroborated the expression patterns of these genes, supporting their involvement in key pathophysiological processes—such as inflammation, iron metabolism, oxidative stress, and tissue remodeling—across different phases of PTE.

## Evaluation of drug efficacy based on PTE marker signals

Based on the aforementioned results, we preliminarily determined that day 7 post-PTE represented the peak period for NETs formation; therefore, this time point was selected for thrombolytic therapy. Combination therapy with rt-PA and DNase1 demonstrated differential efficacy compared to rt-PA monotherapy across multiple key indicators. Histopathological analysis revealed that the reduction in the collagen fiber area fraction in lung tissue was significantly greater in the combination therapy group than in the rt-PA monotherapy group ($P<0.01$), suggesting superior efficacy in alleviating pulmonary fibrosis (Fig 6A). Regarding pulmonary hemodynamics and function, the improvement rates for both PAT ($P<0.01$) and SpO2 ($P<0.001$) were significantly higher in the combination therapy group compared to the monotherapy group (Fig 6B and 6C), indicating that the combination therapy might more effectively ameliorate pulmonary hypertension and pulmonary ventilation function. Concurrently, the improvement rate of LVFS was significantly greater in the combination therapy group than in the rt-PA monotherapy group ($P<0.05$). While LVEF increased significantly in both groups, the improvement rate did not differ significantly between them (Fig 6D and 6E). Furthermore, regarding serological markers related to inflammation and coagulation, both treatments significantly decreased serum levels of MPO-DNA, vWF, BNP, PAI-1, and t-PA. Among these, the recovery rates for vWF and BNP were significantly better in the combination therapy group compared to the rt-PA group (Fig 6F–6J), indicating a more pronounced advantage of the combination therapy in modulating coagulation function and reducing cardiac load. Finally, serum VEGF concentrations showed no significant changes before and after treatment in either group ($P<0.05$, Fig 6K). In summary, within the context of advancing PTE treatment, the identification of novel biomarkers and their guidance for the combined application of rt-PA and DNase1 in PTE patients may offer advantages over rt-PA monotherapy. These benefits are reflected in the amelioration of the degree of pulmonary fibrosis, pulmonary hemodynamic parameters, certain cardiac function indices, and the regulation of specific inflammatory and coagulation markers.

## Additional pharmacological validation supports a NETs-TLR9-NF-kB/GPX4 axis

To provide mechanistic support beyond associative transcriptomic findings, we performed an additional four-group pharmacological validation experiment at day 7 post-PTE (Fig 7). Compared with the control group, the model group

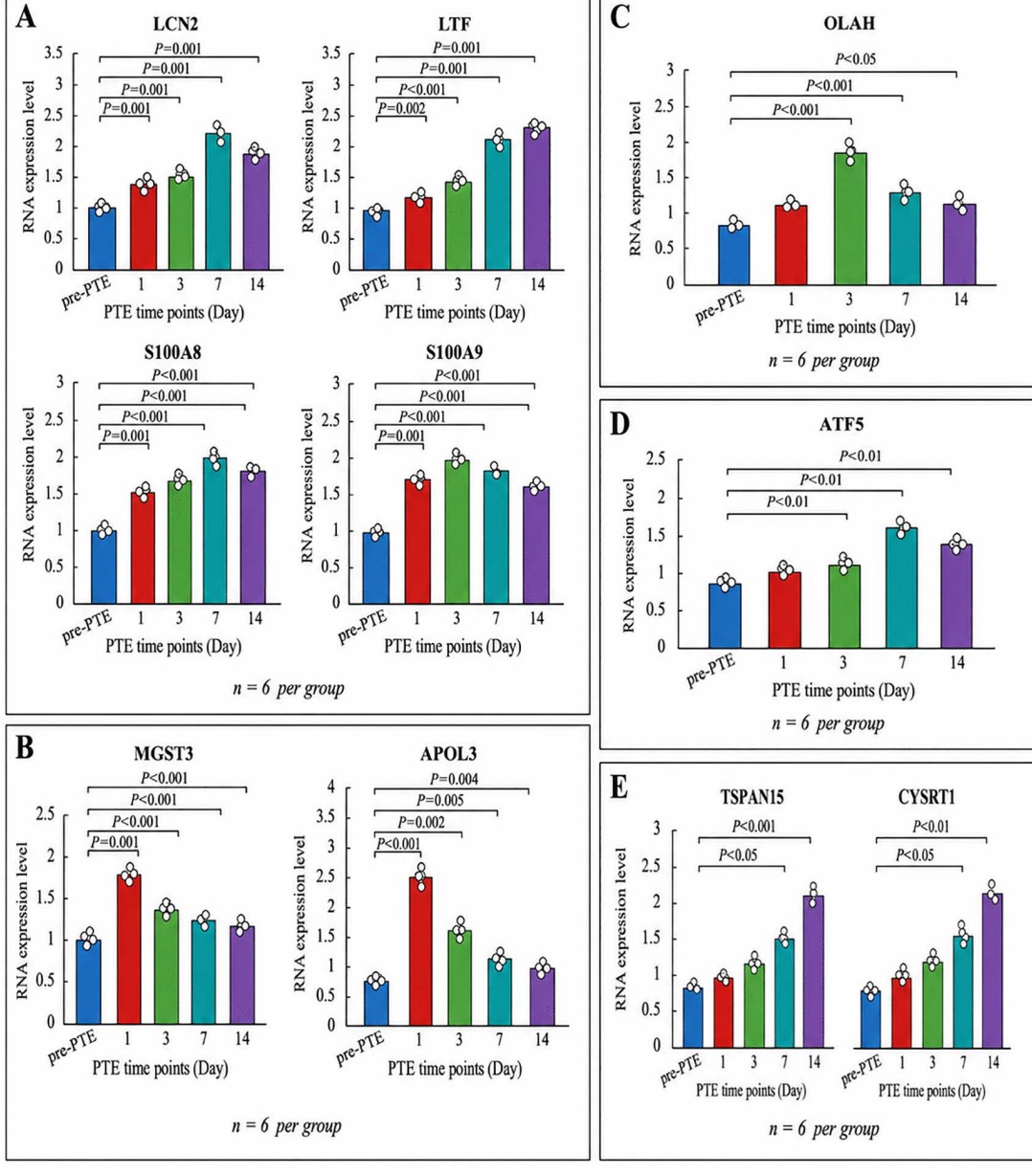

**Fig 5. qRT-PCR validation of common NETs/ferroptosis-related and temporal-specific DEGs after PTE. (A)** Common NETs/ferroptosis-related DEGs at various time points post-PTE modeling. **(B)** Specific DEGs at day 1 post-PTE modeling. **(C)** Specific DEGs at day 3 post-PTE modeling. **(D)** Specific DEGs at day 7 post-PTE modeling. **(E)** Specific DEGs at day 14 post-PTE modeling. n = 6 biologically independent animals per group/time point. *, $P < 0.05$; **, $P < 0.01$; ***, $P < 0.001$.

exhibited markedly higher serum MPO-DNA complexes (used as a surrogate marker of NETs-associated activity), TLR9, phospho-p65, and phospho-p65/total p65 ratio, whereas total p65 abundance remained unchanged. In parallel, GPX4 protein and mRNA expression were significantly reduced, and TLR9 mRNA was increased in the model group. DNase I significantly reduced circulating MPO-DNA complexes and attenuated the increases in TLR9 and phospho-p65 signaling, accompanied by partial restoration of GPX4. Fer-1 did not directly target upstream NETs-associated activity, but it

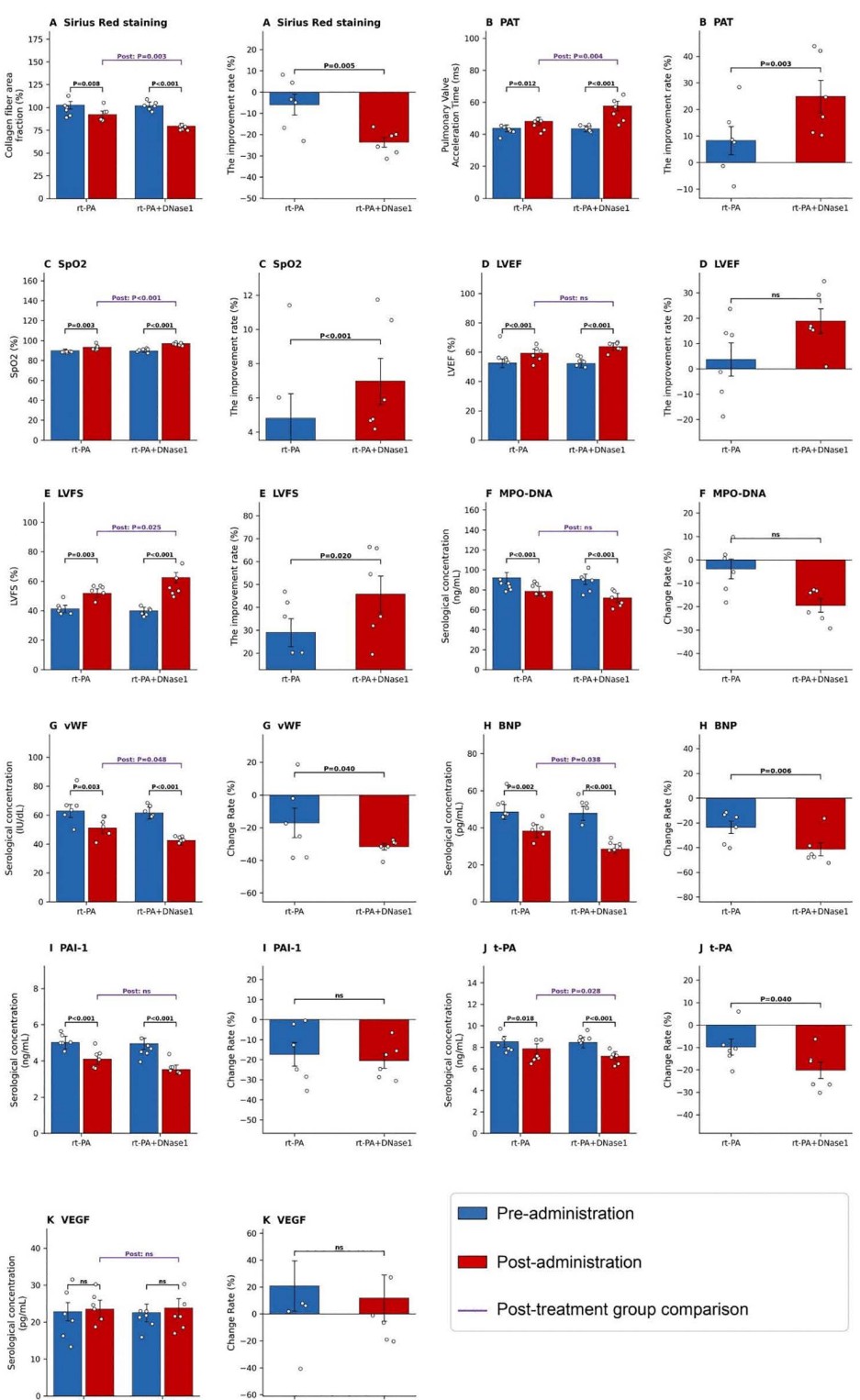

**Fig 6. Changes and improvement rates of various indicators following rt-PA monotherapy and rt-PA+DNase I combination therapy on day 7 after PTE modeling.** **(A)** Collagen fiber area fraction. **(B)** PAT. **(C)** SpO2. **(D)** LVEF. **(E)** LVFS. **(F)** MPO-DNA. **(G)** vWF. **(H)** BNP. **(I)** PAI-1. **(J)** t-PA. **(K)** VEGF. n = 6 biologically independent animals per treatment group. *, $P < 0.05$; **, $P < 0.01$; ***, $P < 0.001$.

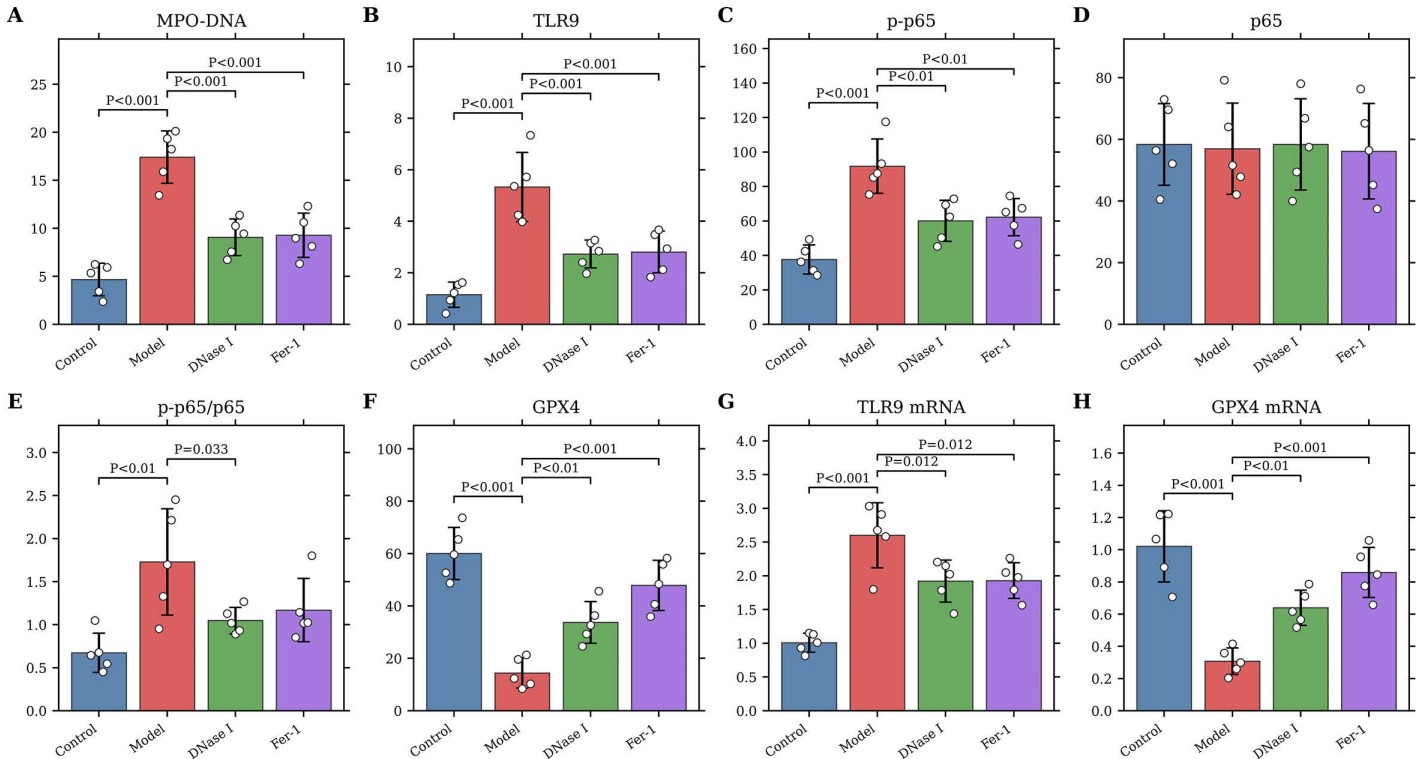

**Fig 7. Additional pharmacological validation of the NETs-TLR9-NF-kB/GPX4 axis in the rabbit PTE model.** Serum levels of **(A)** MPO-DNA, **(B)** TLR9, **(C)** phospho-p65, **(D)** total p65, **(E)** phospho-p65/total p65 ratio, and **(F)** GPX4 were measured by ELISA. Relative mRNA expression of **(G)** TLR9 and **(H)** GPX4 was measured by qRT-PCR. Data are presented as mean ± SD with all individual data points shown; n = 5 biologically independent animals per group. One-way ANOVA followed by Dunnett's multiple-comparisons test was used, with the Model group as the reference. * $P < 0.05$; ** $P < 0.01$; *** $P < 0.001$ versus Model.

also reduced the phospho-p65 response and more prominently restored GPX4 expression, supporting ferroptosis as a downstream effector of injury. Together, these data provide additional pharmacological support for a potential link between NETs-associated activity and the TLR9-NF-kB/GPX4 pathway in PTE-associated injury, while indicating that further tissue-level validation is still required.

## Discussion

In this study, we established a rabbit model of pulmonary thromboembolism to longitudinally investigate the dynamic pathophysiological changes from the acute phase to the subacute phase. Our whole-blood transcriptomic and serological findings provide descriptive insights into the molecular landscape of PTE, with particular emphasis on the sustained activation of NETosis- and ferroptosis-associated transcriptional programs. We also identified temporally associated candidate gene signatures and showed that adjunctive DNase I enhanced rt-PA thrombolytic efficacy in the subacute phase of this preclinical model.

The successfully established animal model of PTE in this study demonstrated characteristics highly consistent with the progression of clinical PTE, both in histopathological and serum biomarker analyses. Histological examination revealed acute inflammatory cell infiltration and alveolar structure disruption in lung tissues on day 1 post-embolization, which progressively developed into fibrotic lesions characterized by significant collagen deposition. This was clearly confirmed

by Masson and Sirius Red staining results on days 7 and 14. This evolving process closely mimics the critical pathological progression of thrombus organization and fibrosis from the acute phase in clinical PTE [9–11], which serves as an important foundation for the development of chronic thromboembolic pulmonary hypertension (CTEPH) [12].

The dynamic changes in serum biomarkers provided strong functional support for the aforementioned histological alterations. The collagen-specific chaperone HSP-47 was significantly elevated on days 3 and 7, suggesting early collagen fibrotic activation and indicating a potential time window for clinical intervention [13]. MPO, a key component of NETs through its incorporation into MPO–DNA complexes, contributes to thrombogenesis by facilitating chromatin decondensation and inducing oxidative modifications. These processes collectively enhance platelet activation and amplify the coagulation cascade in immunothrombotic milieus [14,15]. MPO-DNA levels peaked on day 7 and remained highly expressed until day 14, indicating that NETs may not only participate in acute thrombus formation but also play an important role in the subacute phase by sustaining inflammatory responses and promoting thrombus organization. vWF, a marker of endothelial injury [16], showed continuously increasing levels post-embolization, peaking on day 14. Meanwhile, BNP exhibited an acute rise on day 3 and remained elevated above baseline thereafter. The combined profiles of vWF and BNP indicate that PTE induced sustained pulmonary vascular endothelial dysfunction and increased right ventricular pressure load [17–19], which constitute core mechanisms in the development of pulmonary hypertension [20]. Furthermore, this study observed a critical imbalance in the fibrinolytic system during PTE progression. t-PA showed a transient compensatory increase on day 1 post-embolism, followed by a significant decline on days 7 and 14. In contrast, its key inhibitor, PAI-1, increased continuously and peaked on day 14. t-PA is a critical activator of the fibrinolytic system, primarily catalyzes the conversion of plasminogen to plasmin, promotes fibrin degradation, inhibits excessive thrombus accumulation, and thereby maintains intravascular homeostasis [21]. PAI-1 acts as the primary negative regulator of t-PA, inhibiting t-PA-mediated plasmin generation by forming stable complexes, leading to localized inhibition of fibrinolysis and enhanced thrombus stability, particularly evident in the platelet-rich head region of arterial-like thrombi [21]. In this study, the severe imbalance in the t-PA/PAI-1 ratio likely contributed to significantly suppressed systemic fibrinolytic capacity in the later stages of the disease. D-dimer, a specific product of cross-linked fibrin degradation, originates mainly from the secondary fibrinolysis mediated by plasmin. It reflects the dynamic balance following coagulation activation and fibrin formation, thereby serving as an indirect marker of the presence and extent of thrombogenesis [22]. Its peak level on day 7, rather than during the acute phase on day 1, may reflect a state of ongoing thrombogenesis coexisting with ineffective fibrinolysis at this stage. This progressive decline in fibrinolytic capacity may be a critical reason why thrombi are not completely cleared and ultimately progress to organization and fibrosis.

Our whole-blood transcriptomic profiling offers deeper molecular insights into the evolving pathophysiology of PTE, revealing temporally distinct gene expression patterns that align closely with serological and histological observations. By day 1, pathway enrichment analysis indicated a pronounced systemic inflammatory response, characterized by prominent activation of "regulation of innate immune response" and "response to virus." Notably, significant enrichment was also observed in "ferroptosis," "iron ion homeostasis," and "heme metabolic process," collectively pointing to an early and profound disruption of cellular homeostasis involving inflammation, iron dysregulation, and lipid peroxidation-associated cell death. During day 3 and day 7, transcriptomic shifts revealed sustained dysregulation in "iron ion homeostasis" and "heme/porphyrin biosynthetic processes," accompanied by persistent enrichment of terms associated with "secretory granules" (specific, tertiary, azurophil). These findings suggest that although the initial inflammatory surge subsides, neutrophil degranulation and iron-related metabolic pathways remain highly active. A pivotal transition was evident by Day 14, marking the onset of a chronic remodeling phase. Upregulated DEGs at this stage were strongly enriched in processes including "collagen metabolic process," "extracellular structure organization," "tissue remodeling," and "anatomical structure homeostasis," consistent with histological evidence of progressive collagen deposition. Moreover, sustained enrichment of the "complement and coagulation cascades" corroborated the suppressed fibrinolytic state reflected by elevated PAI-1 and suppressed t-PA levels. Across all time points, the persistent enrichment of "granule-related cellular components" and "iron

transport-related terms" underscores the continuing involvement of neutrophil activity and iron dyshomeostasis throughout the 14-day course, providing a transcriptomic basis for screening for PTE-specific molecules.

A core finding of this study is that transcriptomic analysis combined with qRT-PCR validation supports sustained activation of NETs-associated inflammatory signaling and ferroptosis-related programs during the 14-day course after embolization, rather than restricting these processes to the acute phase alone. Analysis of co-upregulated differentially expressed genes in the whole-blood transcriptome identified 46 genes with sustained high expression. From these, we further identified and validated four key molecules: S100A8, S100A9, LCN2, and LTF. S100A8 and S100A9 can form the calprotectin complex, which serves as a potent mediator of neutrophil and monocyte activation. Their sustained high expression is consistent with prolonged NETs-associated inflammatory activity throughout the disease course [23–25]. Notably, LCN2 and LTF, key regulators of iron metabolism, were identified as genes concurrently associated with both NETs and ferroptosis pathways. These whole-blood findings suggested a mechanistic connection between NETs-associated inflammation and ferroptosis, but they did not by themselves establish a tissue-resolved causal axis.

This study also identified a series of time-specific molecular biomarkers that may reflect key pathological features of PTE at different developmental stages. In the acute phase (Day 1 post-PTE), the expression of MGST3 and APOL3 was significantly upregulated. MGST3, a member of the membrane-associated proteins in eicosanoid and glutathione metabolism (MAPEG) family, primarily functions in cellular detoxification by catalyzing the conjugation of reduced glutathione to electrophilic substrates and exhibiting glutathione peroxidase activity toward lipid hydroperoxides [26]. APOL3 is implicated in apoptosis and cellular membrane injury processes [27]. These changes collectively suggest that oxidative damage and cellular stress dominate the early phase of PTE. By Day 3 post-PTE, OLAH expression peaked. As a key enzyme in fatty acid biosynthesis, OLAH participates in cellular stress responses and inflammatory cascades in various diseases by modulating lipid metabolism pathways [28], which aligns with the active state of the ferroptosis pathway during this stage. In the subacute phase (Day 7 post-PTE), ATF5 expression was markedly increased. As a key regulator of the integrated stress response (ISR), its aberrant expression indicates that cells are initiating adaptive responses or apoptotic programs under sustained microenvironmental pressures, such as hypoxia, inflammation, or endoplasmic reticulum stress [29,30]. By the late subacute phase (Day 14 post-PTE), TSPAN15 and CYSRT1 exhibited progressive upregulation. These genes are involved in cell adhesion, migration [31], and tissue remodeling processes [32]; their increased expression may indicate that thrombus organization, vascular wall remodeling, and fibrosis processes are stabilizing. The identification of these temporally associated candidate markers provides a preliminary basis for the future exploration of potential molecular tools for PTE staging and prognostic assessment, pending validation in independent and larger cohorts.

Both serum MPO-DNA levels and transcriptomic data indicated that day 7 post-PTE represented the peak period of NETs formation, leading us to select this time point for therapeutic intervention. NETs, composed of a DNA scaffold, histones, and granular enzymes (such as NE and MPO), form a dense mesh-like structure that entraps blood cells and cross-links with fibrin, thereby enhancing the mechanical stability of thrombi and potentially limiting rt-PA penetration [33–34]. To move beyond associative evidence, we added an independent pharmacological validation cohort at day 7 post-PTE. In this experiment, the model group showed higher circulating MPO-DNA complexes (used as a surrogate marker of NETs-associated activity), TLR9, phospho-p65, and phospho-p65/total p65 ratio, together with lower GPX4 expression, whereas total p65 abundance remained unchanged. DNase I reduced circulating MPO-DNA levels and was accompanied by attenuation of TLR9–NF-kB signaling and partial restoration of GPX4, suggesting a potential upstream role for NETs-associated activity in this pathway. Fer-1, used as a ferroptosis inhibitor, more prominently restored GPX4 and also dampened downstream phospho-p65 signaling, supporting ferroptosis as a downstream effector process rather than an isolated bystander event. Taken together, these data provide additional pharmacological support for a potential link between NETs-associated activity and the TLR9–NF-kB/GPX4 pathway in PTE-associated injury, while still warranting cautious interpretation.

The rabbit autologous thrombus PTE model used in this study has several advantages over alternative in vivo models. Compared to microsphere embolization models, which produce foreign body-induced inflammation rather than true thrombus-vessel wall interactions, autologous thrombus models more closely recapitulate the natural composition and organization of clinical thrombi, including fibrin networks, entrapped blood cells, and immune cell infiltration. In contrast to surgical ligation models of chronic thromboembolic pulmonary hypertension, our model allows the study of the dynamic transition from acute to subacute PTE. However, this model also has limitations: the bolus injection of pre-formed thrombi does not fully replicate the in situ thrombogenesis observed in clinical PTE, and the rabbit coagulation system differs from that of humans in certain aspects, including platelet function and fibrinolytic activity.

This study also has several limitations. First, although we added an independent pharmacological validation experiment, direct tissue-level visualization of NETs within pulmonary thrombi and in situ assessment of ferroptosis markers in lung tissue were not available in the current revision; therefore, the proposed NETs-TLR9-NF-kB/GPX4 axis should be interpreted as supported rather than definitively proven. Second, although the rabbit model recapitulates key pathological processes of PTE, interspecies differences in physiological and genetic backgrounds between rabbits and humans remain. Third, the transcriptomic analysis was performed using whole blood samples. While this enhances clinical translatability, it does not allow precise delineation of cell type-specific contributions, such as those from neutrophils, monocytes, or endothelial cells. Future studies should incorporate tissue-level staining and single-cell approaches to resolve these distinct cellular responses. Finally, this work primarily focused on the evolution of PTE within 14 days; longer-term observations spanning several months would provide a more comprehensive understanding of progression toward CTEPH.

Additionally, ferroptosis in this study was inferred solely from transcriptomic pathway enrichment and gene expression patterns, without direct measurements of lipid peroxidation, iron-dependent cell death, or GPX4 activity at the tissue or cellular level. Similarly, NETs were primarily identified through circulating MPO-DNA levels and gene signatures, without direct visualization or quantification within thrombi or lung tissue by immunofluorescence for citrullinated histones. Therefore, our findings regarding NETs and ferroptosis should be interpreted as activation of related transcriptional programs rather than confirmed biological events. Furthermore, the intervention study included only two treatment groups (rt-PA alone and rt-PA plus DNase I) without a DNase I-alone control, which limits our ability to determine whether the observed benefits represent additive or synergistic effects. The small sample size (n = 6 per group) also restricts the statistical power for the number of endpoints assessed. Finally, the poor correlation between transcriptome and proteome in terminally differentiated neutrophils should be considered when interpreting our whole-blood transcriptomic data.

## Conclusion

In summary, this rabbit PTE study defines dynamic histological, serological, and transcriptomic changes across the first 14 days after embolization and identifies day 7 as a biologically active window characterized by prominent NETs-associated signaling. The additional DNase I and Fer-1 validation data support the involvement of a NETs-TLR9-NF-kB/GPX4-ferroptosis pathway in PTE-associated injury and reinforce the therapeutic potential of NETs-targeted adjunctive treatment. Further tissue-level studies are needed to confirm the proposed mechanism directly.

## Supporting information

**S1 File. Random forestTOP20 genes.**
(ZIP)

**S1 Text. Gene sequence accession number.**
(DOCX)

**S1 Table. Primer design.**
(XLSX)

## Author contributions

**Conceptualization:** Jialun Chen.

**Data curation:** Lingshan Chao.

**Formal analysis:** Siqin Han.

**Funding acquisition:** Xixin Yan.

**Investigation:** Zaixing Jia.

**Methodology:** Xixin Yan.

**Project administration:** Jialun Chen.

**Resources:** Xixin Yan.

**Software:** Weihua Chen.

**Supervision:** Zhenwei Liu.

**Validation:** Jingwen Li.

**Visualization:** Xixin Yan.

**Writing – original draft:** Jialun Chen.

**Writing – review & editing:** Jialun Chen.

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
