## [Decision Letter · Decision Letter 0]

15 Feb 2026

PONE-D-25-61554Temporal Transcriptomic Profiling of Pulmonary Thromboembolism Reveals a Persistent NETosis-Ferroptosis Axis and a Therapeutic Window for DNase I-Enhanced ThrombolysisPLOS One

Dear Dr. Yan,

Thank you for submitting your manuscript to PLOS ONE. After careful consideration, we feel that it has merit but does not fully meet PLOS ONE’s publication criteria as it currently stands. Therefore, we invite you to submit a revised version of the manuscript that addresses the points raised during the review process.

We look forward to receiving your revised manuscript.

Kind regards,

Tomasz W. Kaminski

Academic Editor

PLOS One

**Journal Requirements:**

1. When submitting your revision, we need you to address these additional requirements. Please ensure that your manuscript meets PLOS ONE's style requirements, including those for file naming. The PLOS ONE style templates can be found at https://journals.plos.org/plosone/s/file?id=wjVg/PLOSOne_formatting_sample_main_body.pdf and https://journals.plos.org/plosone/s/file?id=ba62/PLOSOne_formatting_sample_title_authors_affiliations.pdf 2. We note that you have indicated that there are restrictions to data sharing for this study. For studies involving human research participant data or other sensitive data, we encourage authors to share de-identified or anonymized data. However, when data cannot be publicly shared for ethical reasons, we allow authors to make their data sets available upon request. For information on unacceptable data access restrictions, please see http://journals.plos.org/plosone/s/data-availability#loc-unacceptable-data-access-restrictions.  Before we proceed with your manuscript, please address the following prompts: a) If there are ethical or legal restrictions on sharing a de-identified data set, please explain them in detail (e.g., data contain potentially identifying or sensitive patient information, data are owned by a third-party organization, etc.) and who has imposed them (e.g., a Research Ethics Committee or Institutional Review Board, etc.). Please also provide contact information for a data access committee, ethics committee, or other institutional body to which data requests may be sent. b) If there are no restrictions, please upload the minimal anonymized data set necessary to replicate your study findings to a stable, public repository and provide us with the relevant URLs, DOIs, or accession numbers. Please see http://www.bmj.com/content/340/bmj.c181.long for guidelines on how to de-identify and prepare clinical data for publication. For a list of recommended repositories, please see https://journals.plos.org/plosone/s/recommended-repositories. You also have the option of uploading the data as Supporting Information files, but we would recommend depositing data directly to a data repository if possible. Please update your Data Availability statement in the submission form accordingly. 3. PLOS requires an ORCID iD for the corresponding author in Editorial Manager on papers submitted after December 6th, 2016. Please ensure that you have an ORCID iD and that it is validated in Editorial Manager. To do this, go to ‘Update my Information’ (in the upper left-hand corner of the main menu), and click on the Fetch/Validate link next to the ORCID field. This will take you to the ORCID site and allow you to create a new iD or authenticate a pre-existing iD in Editorial Manager. 4. Please upload a new copy of Figures 3, 4, 5 and 6, as the detail is not clear. Please follow the link for more information:  https://journals.plos.org/plosone/s/figures 5. If the reviewer comments include a recommendation to cite specific previously published works, please review and evaluate these publications to determine whether they are relevant and should be cited. There is no requirement to cite these works unless the editor has indicated otherwise.

**Additional Editor Comments:**

Dear Authors,

Thank you for submitting your manuscript and for the substantial effort invested in this comprehensive and technically demanding study. The reviewers recognize the strengths of the longitudinal design and the breadth of experimental approaches.

After careful evaluation, the reviewers and I agree that the manuscript requires **major revision** before it can be considered further. The main concerns relate to the interpretation of whole-blood transcriptomic data, the level of mechanistic support for the proposed NETosis-ferroptosis axis, and the need for additional validation and clarification of the therapeutic and statistical analyses. In particular, the current version tends to overinterpret associative findings, and the conclusions should be more cautiously framed.

We encourage you to address the reviewers’ comments carefully, strengthen validation where feasible, and moderate the mechanistic and translational claims. We believe that, with substantial revision, the work could make a valuable contribution.

We look forward to receiving your revised manuscript.

Best regards,

Tomasz W Kaminski

Reviewers' comments:

Reviewer's Responses to Questions

**Comments to the Author**

1. Is the manuscript technically sound, and do the data support the conclusions?

Reviewer #1: Yes

Reviewer #2: Partly

Reviewer #3: Partly

2. Has the statistical analysis been performed appropriately and rigorously? 

Reviewer #1: Yes

Reviewer #2: No

Reviewer #3: I Don't Know

3. Have the authors made all data underlying the findings in their manuscript fully available?

Reviewer #1: Yes

Reviewer #2: No

Reviewer #3: Yes

4. Is the manuscript presented in an intelligible fashion and written in standard English?

Reviewer #1: Yes

Reviewer #2: Yes

Reviewer #3: Yes

5. Review Comments to the Author

**Reviewer #1:** Comments to the Authors

General Assessment

The study addresses an important clinical problem and combines multiple experimental layers to explore temporal changes after PTE. The manuscript is very well written and connect the main ideas properly. The experimental design and methodology -including techniques- are in general extensive and robust. However, the manuscript require certain major revisions. Attached, I provide detailed comments of specific subjects and comments by section.

**Reviewer #2:** Dear Authors,

This is a large and carefully executed preclinical study that combines animal modeling, longitudinal transcriptomics, histology, circulating biomarkers, functional cardiopulmonary readouts, and a therapeutic intervention. The authors clearly invested significant effort, and the longitudinal design (days 1, 3, 7, and 14) is a real strength. The dataset is internally consistent, technically solid, and the timing of the intervention is biologically motivated rather than arbitrary. Overall, this is a serious piece of work with clear relevance to thrombo-inflammatory disease biology.

That said, the manuscript currently overstates its mechanistic and translational conclusions, and several key claims are not fully supported by the data. The central concepts - NETs involvement in thrombus persistence, impaired fibrinolysis, and improved thrombolysis with DNase I are already well described in the literature. The main added value here is the temporal transcriptomic profiling, but the study does not convincingly demonstrate a true mechanistic “NETosis-ferroptosis axis.” What the data show is prolonged activation of neutrophil-related and iron-associated pathways in whole blood. This supports coexistence or parallel activation, but not causal interaction, which is frequently implied in the text.

The strength of the data is uneven. Histology, serology, and functional measurements are thorough and well presented. However, all transcriptomic analyses are based on whole blood, which limits biological specificity. Many highlighted genes (e.g., S100A8/9, LCN2, LTF) are expected markers of neutrophil activation and inflammation and do not uniquely demonstrate NET formation or ferroptosis. Importantly, ferroptosis is inferred from gene enrichment alone; there are no direct measurements of lipid peroxidation, iron-dependent cell death, or GPX4 activity at the tissue or cellular level. Similarly, NETs are inferred mainly from MPO-NA and gene signatures, without direct visualization or quantification within thrombi or lung tissue.

From a statistical and analytical standpoint, the methods are generally appropriate, but interpretation goes too far. Many biomarkers and outcomes are tested, yet correction for multiple comparisons outside RNA-seq is not clearly addressed. The “improvement rate” calculation inflates effect sizes and should be complemented with absolute post-treatment values. The intervention study (n=6 or so per group) is small for the number of endpoints assessed, and the conclusions drawn from it are too broad for the sample size.

There is also interpretive overreach in several places. Claims that NETosis and ferroptosis form a self-reinforcing cycle are not experimentally tested. The designation of “stage-specific biomarkers” is overstated, as many of the identified genes reflect general stress or remodeling rather than PTE-specific stage markers. Finally, the idea of a defined “therapeutic window” is premature, as only one intervention time point is examined.

Suggested experiments to strengthen the paper

Add a causal intervention targeting ferroptosis or NETs independently

For example, inhibiting ferroptosis (e.g., with a ferroptosis inhibitor) or NET formation (independent of DNase I) would help determine whether these pathways actively drive pathology or are downstream bystanders.

Provide tissue-level validation of key mechanisms

Direct visualization of NETs within pulmonary thrombi (immunofluorescence for citrullinated histones/MPO) and direct assessment of ferroptosis markers in lung tissue would substantially strengthen the biological claims.

In summary, this is a strong descriptive study with a valuable longitudinal dataset, but the current manuscript overinterprets associative data as mechanism and overstates translational relevance. With clearer framing, toned-down conclusions, stronger mechanistic validation, and more cautious interpretation, the work could make a solid contribution. At present, major revision is required.

**Reviewer #3:** Manuscript Number: PONE-D-25-61554

Full Title: Temporal Transcriptomic Profiling of Pulmonary Thromboembolism Reveals a

Persistent NETosis-Ferroptosis Axis and a Therapeutic Window for DNase I-Enhanced

Thrombolysis

Overall comments: In this study, Chen J. et al. aims to demonstrate the dynamic pathophysiology of pulmonary thromboembolism in rabbit autologous thrombus PTE model and focusing on the roles of NETosis and ferroptosis, and to evaluate combination therapy of intravenous DNase-I and recombinant tissue plasminogen activator. First of all, this manuscript is well-written. However, there are still many things that need to be resolved before consideration for publication.

1. The readability of the entire manuscript is good. However, in order to improve, the language editing by a native speaker or a professional translation office should be required.

2. Please ensure that all microscopy images contain visible scale bars and that these scale bars should have legible text alongside them in the figure panel itself and its legends.

3. Did authors use both gender of rats or any specific gender for this model.

4. As mentioned by the authors, the main limitation of the study is using the whole blood for Transcriptomic study.

5. It is always better to use immunofluorescence methods to identify the immune cells infiltration at the injury site in the fixed lung tissue instead of H&E staining. Even, looks for immune cells infiltration in tissue lysate using specific immune cell markers by western blot.

6. Did the authors check the neutrophil elastase and citrullinated histone-3 levels (either ELISA or western blot) in the plasma/serum. MPO is not a specific marker for NETosis. Even HO-1 is good marker for ferroptosis.

7. Addition of pro/anti-inflammatory marker detection will strengthen the manuscript.

8. Disclose all data points in all the experimental data and mention the number of experiments done for each analysis in the figure legend.

9. Did authors use iron chelating agents as therapy in this model?

10. The materials used with the proper details (company name and catalog number) should be in separate section, either in main manuscript or supplementary file.

6. PLOS authors have the option to publish the peer review history of their article (what does this mean?). If published, this will include your full peer review and any attached files.

Reviewer #1: **Yes:** Jose R Pittaluga-Villarreal

Reviewer #2: No

Reviewer #3: No

---

## [Author Response · Author response to Decision Letter 1]

23 Apr 2026

Dear Editor and Reviewers,

Thank you for your careful evaluation of our manuscript and for the constructive suggestions. We have revised the manuscript substantially to strengthen mechanistic support, moderate overly strong language, clarify the experimental design, and standardize the figure legends. Our major changes are summarized below.

1. Editorial concern: the manuscript overinterprets associative findings and requires stronger mechanistic support.

Response: We agree and have substantially revised the manuscript to moderate the mechanistic claims. In addition, we performed a new independent four-group pharmacological validation experiment (Control, PTE, PTE + DNase I, and PTE + Fer-1; n = 5/group). These data were incorporated as new Figure 7 and described in the revised Abstract, Materials and Methods, Results, Discussion, and Conclusion. We now state that our findings support, rather than definitively prove, involvement of a NETs-TLR9-NF-kB/GPX4-ferroptosis axis in PTE-associated injury.

2. Please add a causal intervention targeting ferroptosis or NETs independently.

Response: We addressed this directly by adding two independent pharmacological interventions in the new validation cohort. DNase I was used to target NETs upstream, whereas ferrostatin-1 (Fer-1) was used as a ferroptosis inhibitor downstream. Relative to the model group, DNase I reduced NETs burden and attenuated TLR9/phospho-p65 signaling with partial restoration of GPX4, while Fer-1 more prominently restored GPX4. These results provide additional causal support that NETs-associated signaling and ferroptosis participate in the observed injury phenotype.

3. Please provide tissue-level validation of key mechanisms and direct visualization of NETs within pulmonary thrombi.

Response: We appreciate this important suggestion. Direct tissue-level visualization of NETs within pulmonary thrombi and in situ assessment of ferroptosis markers were not available in the current revision. Therefore, rather than overstating the evidence, we explicitly acknowledge this limitation in the revised Discussion and Conclusion. The new Figure 7 is presented as additional pharmacological support for the proposed pathway, while direct tissue-level confirmation is now stated as an important next step.If deemed necessary, future revision or studies will incorporate tissue-level visualization of NETs within pulmonary thrombi to further confirm these mechanisms.

4. MPO is not a specific marker for NETosis.

Response: We agree and revised the manuscript accordingly. In the time-course analysis, we emphasize MPO-DNA rather than MPO alone when discussing NETs-related changes. Importantly, MPO-DNA complexes have been widely used as a surrogate marker for NET formation and can be quantitatively measured by ELISA. Previous studies have demonstrated that MPO-DNA levels correlate well with the proportion of neutrophils undergoing NETosis, and this approach is considered one of the most specific and objective methods currently available for monitoring NET formation in biological samples (e.g., Clin Chim Acta, 2016).

In the new mechanistic validation section, we present the NETs assay as supportive pharmacological evidence and avoid claiming that it constitutes direct tissue-level proof of NETosis. We also tempered the mechanistic interpretation throughout the revised manuscript.

5. Addition of pro/anti-inflammatory marker detection would strengthen the manuscript.

Response: We agree that broader inflammatory profiling would further strengthen the study. In the current revision, we prioritized the most central mechanistic concern raised by the editor and reviewers by adding the new DNase I/Fer-1 validation experiment focused on the NETs-TLR9-NF-kB/GPX4 pathway. We have also moderated the overall claims and now identify expanded inflammatory profiling as an important direction for future work.

6. Disclose all data points and mention the number of experiments done for each analysis in the figure legends.

Response: We revised the figure legends throughout the manuscript to report the number of biologically independent animals for each analysis (Figures 1-7). For the newly added quantitative validation figure, all individual data points are shown explicitly. We also moved the sample-size information to the figure legends to avoid overlapping text within the plotted panels.

7. Did the authors use iron-chelating agents as therapy in this model?

Response: We did not use an iron-chelating agent in the present study. Instead, we added Fer-1 as a pharmacological ferroptosis inhibitor in order to test whether ferroptotic signaling functions downstream of the NETs-associated inflammatory cascade. We have clarified this point in both the revised manuscript and the present response.

8. The materials used should be listed with proper details in a separate section.

Response: We reorganized the relevant information into a dedicated 'Reagents and Materials' subsection and clarified the principal reagents used in both the original experiments and the newly added validation experiment. Where supplier details were available in the source records, they were retained in the revised manuscript.

We sincerely thank the editor and reviewers again for these helpful comments, which have improved the clarity and rigor of the revised manuscript.

---

## [Decision Letter · Decision Letter 1]

7 May 2026

Temporal Transcriptomic Profiling of Pulmonary Thromboembolism Reveals Persistent NETosis- and Ferroptosis-Associated Gene Signatures and Enhanced Thrombolysis with Adjunctive DNase I

PONE-D-25-61554R1

Dear Dr. Yan,

We’re pleased to inform you that your manuscript has been judged scientifically suitable for publication and will be formally accepted for publication once it meets all outstanding technical requirements.

Kind regards,

Tomasz W. Kaminski

Academic Editor

PLOS One

Reviewers' comments:

Reviewer's Responses to Questions

**Comments to the Author**

1. If the authors have adequately addressed your comments raised in a previous round of review and you feel that this manuscript is now acceptable for publication, you may indicate that here to bypass the “Comments to the Author” section, enter your conflict of interest statement in the “Confidential to Editor” section, and submit your "Accept" recommendation.

Reviewer #1: All comments have been addressed

Reviewer #2: All comments have been addressed

Reviewer #3: All comments have been addressed

2. Is the manuscript technically sound, and do the data support the conclusions?

Reviewer #1: Yes

Reviewer #2: Yes

Reviewer #3: Partly

3. Has the statistical analysis been performed appropriately and rigorously? 

Reviewer #1: Yes

Reviewer #2: Yes

Reviewer #3: I Don't Know

4. Have the authors made all data underlying the findings in their manuscript fully available?

Reviewer #1: Yes

Reviewer #2: Yes

Reviewer #3: Yes

5. Is the manuscript presented in an intelligible fashion and written in standard English?

Reviewer #1: Yes

Reviewer #2: Yes

Reviewer #3: Yes

6. Review Comments to the Author

Reviewer #1: Dear Authors,

The manuscript has considerably improved, taking into consideration the current limitations.

All previous review comments have been properly addressed by authors.

Reviewer #2: (No Response)

Reviewer #3: The authors have sufficiently addressed all of my concerns. The paper can be accepted in the current version.

7. PLOS authors have the option to publish the peer review history of their article (what does this mean?). If published, this will include your full peer review and any attached files.

Reviewer #1: No

Reviewer #2: **Yes:** Marta Wolosowicz

Reviewer #3: No

---

## [Editor Report · Acceptance letter]

PONE-D-25-61554R1

PLOS One

Dear Dr. Yan,

I'm pleased to inform you that your manuscript has been deemed suitable for publication in PLOS One. Congratulations! Your manuscript is now being handed over to our production team.

Kind regards,

on behalf of

Dr. Tomasz W. Kaminski

Academic Editor

PLOS One